ecology/environmental science

coral bleaching, mesophotic coral communities, spatial refuge, climate change, French Polynesia

**Author for correspondence:**
Gonzalo Pérez-Rosales
e-mail: gonzalo.prb@gmail.com

†Under The Pole Consortium is listed below the Acknowledgements.
‡Equal contribution as last authors.

# Mesophotic coral communities escape thermal coral bleaching in French Polynesia

Gonzalo Pérez-Rosales[1,2], Héloïse Rouzé[1,2], Gergely Torda[3], Pim Bongaerts[4], Michel Pichon[5], Under The Pole Consortium[6,†], Valeriano Parravicini[2,‡] and Laetitia Hédouin[1,2,‡]

[1]PSL Research University, EPHE-UPVD-CNRS, USR 3278 CRIOBE, BP 1013 Papetoai, 98729 Moorea, French Polynesia
[2]PSL Université Paris: EPHE-UPVD-CNRS, USR 3278 CRIOBE, Université de Perpignan, 52 Avenue Paul Alduy, 66860 Perpignan, France
[3]ARC Centre of Excellence for Coral Reef Studies, James Cook University, Townsville, QLD 4811, Australia
[4]California Academy of Sciences, San Francisco, CA 94118, USA
[5]Biodiversity Section, Queensland Museum, Townsville, QLD 4811, Australia
[6]Under The Pole, 29900 Concarneau, France

GP-R, 0000-0001-6577-3416; HR, 0000-0003-3380-0883; GT, 0000-0002-4585-3045; PB, 0000-0001-6747-6044; VP, 0000-0002-3408-1625; LH, 0000-0001-6565-3633

Climate change and consequent coral bleaching are causing the disappearance of reef-building corals worldwide. While bleaching episodes significantly impact shallow waters, little is known about their impact on mesophotic coral communities. We studied the prevalence of coral bleaching two to three months after a heat stress event, along an extreme depth range from 6 to 90 m in French Polynesia. Bayesian modelling showed a decreasing probability of bleaching of all coral genera over depth, with little to no bleaching observed at lower mesophotic depths (greater than or equal to 60 m). We found that depth-generalist corals benefit more from increasing depth than depth-specialists (corals with a narrow depth range). Our data suggest that the reduced prevalence of bleaching with depth, especially from shallow to upper mesophotic depths (40 m), had a stronger relation with the light-irradiance attenuation than temperature. While acknowledging the geographical and temporal variability of the role of mesophotic reefs as spatial refuges during thermal stress, we ought to understand why coral bleaching reduces with depth. Future studies should consider repeated monitoring and detailed ecophysiological and environmental data. Our study demonstrated how increasing depth may offer a level of protection and that lower mesophotic communities could escape the impacts of a thermal bleaching event.

# 1. Introduction

Warming sea surface temperature leading to spatially and taxonomically widespread bleaching events is one of the major drivers of the loss of reef-building corals [1]. Corals, the habitat engineers of one of the most diverse marine ecosystems of our planet, live in obligate symbiosis with unicellular dinoflagellates from the family Symbiodiniaceae. This symbiosis readily breaks down under unfavourable environmental conditions, most commonly during anomalously high temperature and light-irradiance exposure events [2–4]. With the loss of dinoflagellates from the host tissue, corals are nutritionally compromised and, unless the environmental conditions improve, they die [5,6]. Bleaching susceptibility is taxon- and location-specific [7–10] and mass bleaching events lead to rapid compositional shifts in the benthic community [11]. With the ever-increasing frequency and severity of bleaching events, shallow corals are disappearing at an alarming rate [1,12–14], and the quest to identify thermally tolerant corals and coral reefs is of great interest for conservation and management [15,16].

Because temperature and light attenuate with depth, it has been proposed that mesophotic coral assemblages (i.e. below 30 m; [17]) may act as a spatial refuge for corals during global and local bleaching events [18]. However, despite extensive debate regarding the overlap in community structure [19,20] and connectivity between shallow and mesophotic coral habitats [21,22], there are very few studies that actually assess bleaching well into lower mesophotic depths [23,24]. Here, we address this knowledge gap by conducting scleractinian coral surveys to 90 m depth during a mass bleaching event in French Polynesia in 2019. We test the hypothesis that bleaching impacts decrease with depth; assess taxonomic patterns in bleaching susceptibility over depth and determine whether a decrease in bleaching impacts over depth leaves deeper mesophotic communities unaffected.

# 2. Material and methods

## 2.1. Study locations and sampling protocol

Benthic bleaching surveys were conducted at two locations in French Polynesia: Moorea (Society Archipelago: 17°28.631′ S, 149°51.067′ W) and Makatea (Tuamotu Archipelago: 15°49.383′ S, 148°16.758′ W) from 6 to 90 m depth (electronic supplementary material, figure S1A). The forereefs of the north coast of Moorea are characterized by a gentle slope (40–60°) from the surface down to 70–80 m depth, where it drops almost vertically. The reefs of the northwest side of Makatea are characterized by a very gentle upper slope (less than 10°) until about 10 m depth, where it drops off almost vertically. Coral bleaching surveys were done once at each location on the 14 and 15 June 2019 in Makatea and 12 and 13 July 2019 in Moorea, two to three months after the peak in sea surface temperature that triggered a mass bleaching event in March–April 2019. Thirty photo-quadrats (0.75 × 0.75 m) were randomly taken at each of five different depths (6, 20, 40, 60 and 90 m; a total of 150 photos at each location), with constant sampling effort at each depth, covering around 17 m$^2$ per isobath and 85 m$^2$ per location (figure 1a). Deep dives were done by the Under the Pole team using mixed-gas and closed-circuit rebreathers. Quadrat photos were white balance corrected using a white reference plate attached to each quadrat.

## 2.2. Environmental parameters

We characterized the temperature regime that induced bleaching at our sites using different techniques. First, we computed the cumulative heat stress as degree heating weeks (DHW) from NOAA Satellite Sea Surface Temperatures [25,26]. Second, because satellite measurements are relevant only for shallow waters, we also deployed *in situ* temperature (HOBO Water Temperature Pro v. 2 Data) and light (DEFI2-L JFE Advantech) loggers that measured light for photosynthetically active radiation (PAR) at the different sampling depths. A detailed explanation of how these data were collected, normalized according to 6 m depth because it was the departing reference for loggers and bleaching assessments and processed using the Beer–Lambert equation [27,28] are available in electronic supplementary material, figures S2 and S3.

## 2.3. Coral bleaching classification and statistical analysis

We were interested in exploring the relationship between depth and the likelihood of corals to bleach. In each photo-quadrat, we counted the number of colonies larger than 5 cm [29] and identified them to the

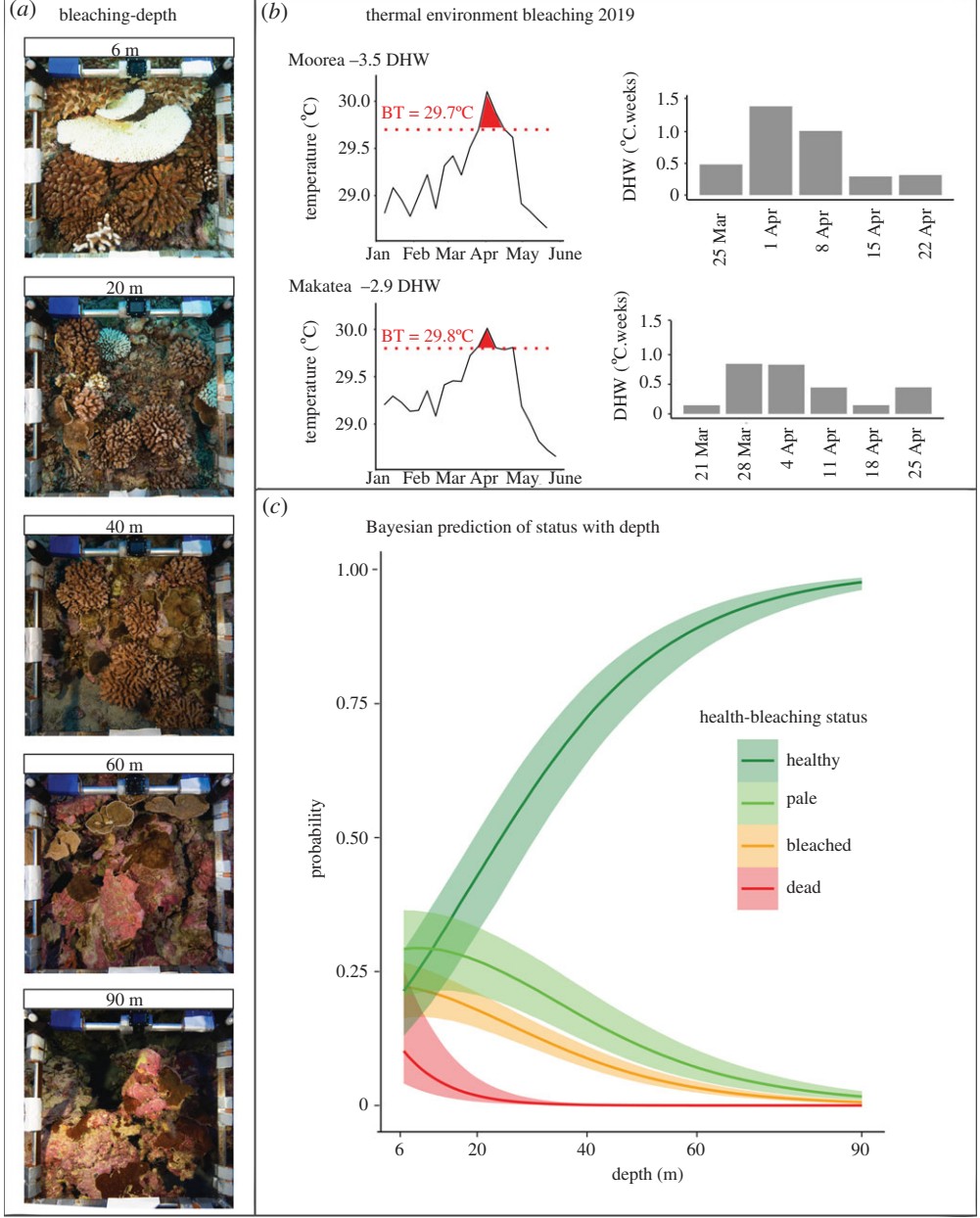

**Figure 1.** (*a*) Sample photo-quadrats at different depths. (*b*) Thermal environment with degree heating weeks (DHW) values using Satellite Coral Reef Watch Temperature. (Left) Sea surface temperatures during the bleaching episode. Red fill colour shows the exposure period above the bleaching threshold marked with a dotted line. (Right) Weekly development of the DHW. (*c*) Bayesian prediction probability of the likelihood of corals to be healthy, pale, bleached or dead as a function of depth.

highest taxonomic resolution possible (i.e. at least genus level and, in several cases, species level). We classified each colony using reference colour cards [30]. Although natural changes of coloration can vary with depth, we considered as references the previous healthy colours observed in the photo-quadrats of the DEEPHOPE expedition [31]. Coral colonies were considered 'healthy' when fully pigmented with their normal colour; 'pale' when colonies had lost part of their pigmentation, but Symbiodiniaceae were still visible in their tissues; 'bleached' when completely white showing their skeleton through the tissue; and 'recently dead' when no living tissues remained over the skeleton, but still had discernible skeletal structure even if colonies started to be covered by turfing algae to varying extents. By contrast, colonies overgrown by thick and dark turf, crustose coralline algae and boring organisms were excluded from the analysis as their mortality probably predates the studied thermal coral bleaching event. Finally, when colonies were patchily pigmented, they were classified by the most severely bleached patch (e.g. a colony 25% bleached and 75% pale was considered as

bleached and colonies 30% pale and 70% healthy as pale). To prevent potential observer bias, all photo-quadrats were analysed and scored by a single observer who had been previously trained with the analysis of 2880 quadrats from the DEEPHOPE expedition [31].

To explore the relationship between the likelihood of bleaching and depth, we fitted a Bayesian model in R, using the brms package [32] with a multinomial logit link function, a random intercept for the 'location' factor and a random intercept and a random slope for the 'genus' factor. We used uninformative priors and ran the model for four MCMC chains using 7000 iterations per chain after 3000 iterations of warm-up. We used the posterior likelihood of being healthy at the shallowest occurrence of each genus as a benchmark to account for genus-specific bleaching resistance. Similarly, for each genus, we calculated the posterior likelihood of being healthy at their deepest occurrence to evaluate depth-driven health benefits. All analyses were performed in RStudio. Data and codes are available in https://github.com/gonzaloprb/Deep_Bleaching_French_Polynesia.

# 3. Results

## 3.1. Environmental characteristics

Both locations experienced heat stress with sea temperatures above their bleaching threshold (BT). Peak sea surface temperature anomalies occurred at the end of March–April 2019 both in Moorea and Makatea. During the bleaching event, Moorea experienced a cumulative heat stress of 3.5 DHW and Makatea of 2.9 DHW (figure 1b). During the bleaching survey, sea surface temperatures had decreased below the BT. Temperatures were similar at both locations, with circadian variations of 0.3°C down to 40 m and high variability below 60 m (electronic supplementary material, figure S2). Vertical profiles revealed that temperatures slightly decreased or were constant from the surface to 60 m depth (greater than 95% of surface temperatures). These temperature values were consistent with the vertical conductivity, temperature and depth (CTD) profile in Makatea (Pearson = 0.95, p-value = 0.0007) (electronic supplementary material, figure S3). Light decreased exponentially with increasing depth. Relative to the value at 6 m, PAR was about 25% at 40 m and only 4% at 90 m. These light values were also consistent with the results obtained from a vertical CTD profile (PAR was less than 20% of the surface light at 40 m and less than 3% at 90 m; Pearson = 0.99, p-value = $4.8 \times 10^{-7}$) and well fitted with the Beer–Lambert equation (electronic supplementary material, figures S2 and S3).

## 3.2. Bleaching along the depth gradient

The proportion of colonies impacted by bleaching dramatically declined with depth at both locations (electronic supplementary material, figure S1B). Despite differences between the two locations in the numbers of colonies over depth (e.g. decreasing in Moorea below 40 m, and remaining constant in Makatea with a maximum number at 60 m), they did not significantly affect the proportional measures of coral bleaching with depth. Consistently, our model shows that the overall probability of bleaching across all coral genera, or dying shortly before the survey period, presumably due to bleaching, decreases with depth (figure 1b). At 6 m, the probability for a coral colony to be bleached or healthy is almost identical (0.25), while at 20 m, the probability of bleaching reduces by more than half and the probability of being healthy doubles to 0.44 ± 0.2 standard error estimate (SEE). At 40 m, the probability of being healthy is 0.73 ± 0.2 SEE, and at 90 m, 0.98 ± 0.02 SEE.

Additionally, our model detected a remarkable variation in the bleaching sensitivity among genera (figure 2). Accounting for genus-specific zonation patterns (i.e. upper and lower depth limits of coral genera), we found that corals only present at mesophotic depths were the least sensitive to bleaching, i.e. *Echinophyllia* (0.94 ± 0.05 SEE) and *Pachyseris* 0.91 ± 0.05 SEE). Among the genera present at 6 m, *Porites* (0.51 ± 0.06 SEE) and *Leptoseris* (0.42 ± 0.09 SEE) were the most resistant to bleaching, while *Astrea* (0.2 ± 0.09 SEE), *Acropora* (0.21 ± 0.07 SEE) and *Montipora* (0.21 ± 0.05 SEE) were the most sensitive. Colonies of all coral genera were healthier at their lower depth limits and the benefit of depth was stronger for depth-generalists than for genera with narrow depth ranges. For instance, the probability of a *Pocillopora* colony to be healthy increased by 0.7 from its upper limit at 6 m (0.22 ± 0.02 SEE) to its lower limit at 60 m (0.92 ± 0.03 SEE). By contrast, the probability of an *Astrea* colony only increased by 0.16 from 6 m (0.2 ± 0.09 SEE) to 20 m (0.36 ± 0.11 SEE) (figure 2a). Although the increasing probability of being healthy, and a decreasing probability of suffering bleaching effects with depth was a trend shared by all coral genera, there were genus-specific differences in the benefits

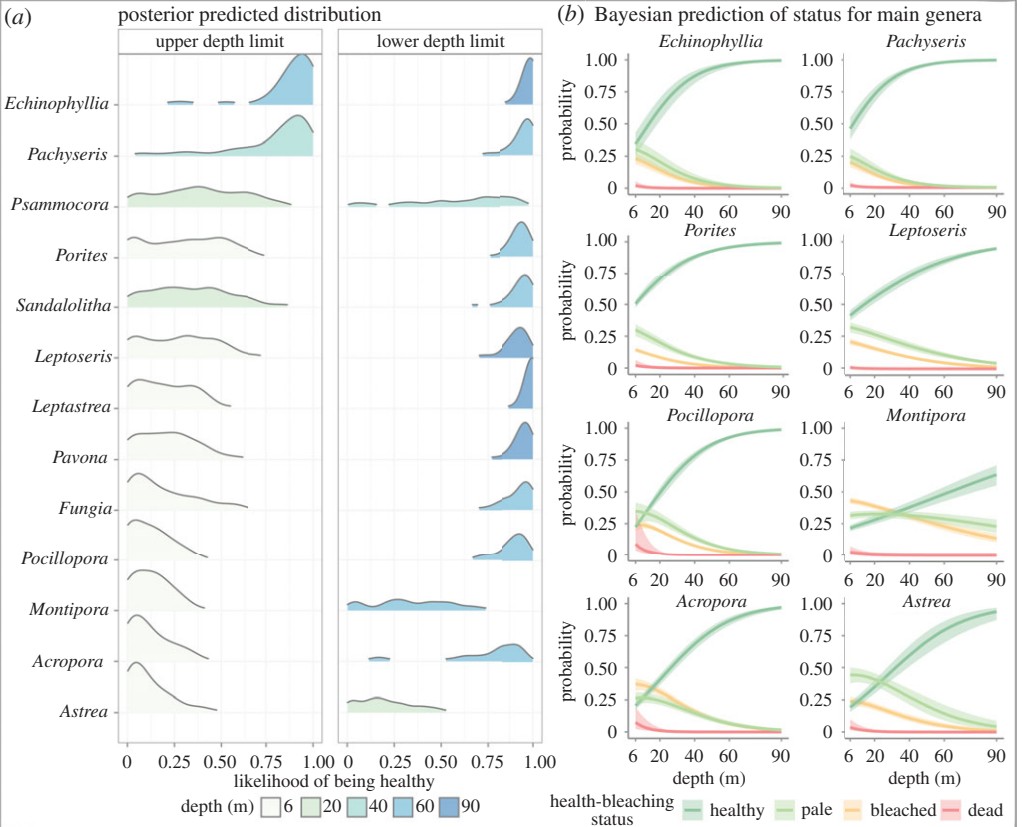

**Figure 2.** (*a*) Bleaching sensitivity of coral genera based on the posterior likelihood of being healthy. Predictions are displayed for the extremes of the depth range specific to each coral genus (i.e. upper and lower depth local limits). Cases with less than 25 replicates per depth and/or less than 100 replicates total were removed. (*b*) Bayesian prediction probability of the likelihood of corals to be healthy, pale, bleached or dead as a function of depth. The most bleaching-tolerant coral genera are *Echinophyllia* and *Pachyseris*. The most bleaching-sensitive genera are *Acropora* and *Astrea*. *Porites*, *Leptoseris*, *Pocillopora* and *Montipora* are the most common genera in the coral assemblages of the surveyed sites.

provided by depth regardless of the extent of the depth range. For instance, despite *Acropora* and *Montipora* having similar depth distributions and thermal sensitivities in shallow waters, *Acropora* benefited from depth more than *Montipora* at 40 m; *Acropora* already had a approximately 0.75 probability of being healthy, while the probability of *Montipora* colonies to be healthy was still less than 0.5 (figure 2*b*).

## 4. Discussion

This study shows a decrease in the incidence of coral bleaching across an extreme depth range (i.e. 6–90 m) in French Polynesia, as measured two to three months after the peak of a heat stress event. Consistent with previous findings, our results suggest that the probability of coral bleaching is reduced with increasing depth [29,33–35]. However, while previous studies were limited to the boundaries of non-technical diving (i.e. 30–40 m depth), we surveyed coral assemblages down to 90 m [23,24,36]. In the upper mesophotic zone (40 m), we found no signs of recent mortality, and bleaching (i.e. bleached or pale) was 3.4-fold less common than in shallow water, in concordance with previous studies [29,33]. At the mid-lower mesophotic zone (60 m) and below, we found no signs of recent mortality and bleaching was virtually absent.

Our Bayesian model predicts taxon-specific sensitivity of corals to bleaching along their respective depth ranges. We show that the reduction in the prevalence of bleached colonies is not simply an effect of taxonomic zonation [37], at least not at genus level. Coral genera inhabiting exclusively the mesophotic zone, from 40 to 90 m, often suffer bleaching at the upper end of their depth range and decrease in bleaching towards their deepest occurrence. While several studies have tried to classify the sensitivity of corals to bleaching [7,8,38–42], few studies have so far explored how bleaching

sensitivity varies across the depth gradient [33,43]. Incorporating random intercept and slope for taxon allowed us to isolate the probability of coral bleaching from the depth distribution of a given genus. Our predicted results corroborated earlier studies that Acroporidae (for the purposes of this study, *Acropora* and *Montipora*) are extremely sensitive to bleaching [41,44], but also showed that *Montipora* benefits less from the effect of depth than *Acropora*. Moreover, we found that higher probability of detecting healthy colonies with increasing depth was a common trend for all genera regardless of their depth distribution (extending the work by Frade *et al*. [33]; but see Crosbie *et al*. [43]). Finally, we found that depth-specialist coral genera were the least sensitive to bleaching; and that the health status of depth-generalists improved more with increasing depth than that of depth-specialists. Such generic patterns could be elaborated in future studies, including the interspecific bleaching variability within genera [45].

The reduction of coral bleaching from the surface to depth can be attributed to changing environmental conditions, the physiological capacities of corals or the interplay of the two [46]. By assessing the impact of bleaching two to three months after the peak of heat stress, we have a snapshot view of the effect of heat stress on corals along the depth gradient. The timing of our surveys does not allow us to differentiate unaffected corals from those that recovered quickly following the peak of the thermal stress [47]. Nevertheless, we reliably identify the most susceptible coral genera to bleaching along an unprecedented depth gradient because corals severely bleached during the heat stress will die or remain bleached for months before the symbionts come back [47,48]. During the survey, sea surface temperatures were below the BT, and it is unlikely to see healthy corals bleach because of temperature without heat stress. Anomalously high sea surface temperatures are usually considered the primary driver of coral bleaching but are not the only abiotic parameter involved [2–4]. For instance, the severe bleaching observed in shallow coral assemblages, despite the relatively low values of DHW [26], was probably due to the interaction between temperature and light irradiation. This interaction could explain why corals bleached, with no mortality, in 2016 at even lower DHW values at some sites of French Polynesia [49–51]. Vertical profiles from the limited environmental data at our study sites showed stable temperatures from the surface to 40 m and significant temporal variability at lower mesophotic depths (60–90 m). This high variability (e.g. up to 1 or 2°C within minutes; electronic supplementary material, figure S2) may provide relief from high temperatures and/or increase organisms' stress tolerance [52–54]. The variability is probably associated with the common internal tidal/gravity waves of the Pacific Islands [55–58], which have proved to reduce heat stress in several locations [54,59,60]. Additionally, the thermocline's depth recorded in our study corresponded to depths reported for the region [57,61,62]. Although both phenomena appear too deep to have an effect at the timepoint of observations, they might have provided extensive heat relief before our measurements. Therefore, more rigorous and long-term studies are necessary to understand the role of oceanographic processes in reducing heat stress on corals and, hence, identify reefs less affected by coral bleaching [42,54,59,60,63,64].

Coral bleaching is a stress response to the increasing concentration of toxic by-products of photosynthesis, most commonly brought about by high temperature, compounded by other factors such as light intensity and water chemistry [3,5,6,65]. By contrast to temperature, and in accordance with the Beer–Lambert law [28], light levels decreased exponentially with depth (electronic supplementary material, figure S2). This reciprocal trend to the likelihood of being healthy suggests that light was a stronger predictor of bleaching than temperature in this study (electronic supplementary material, figure S4). The reduction of PAR with depth [21,66] implies that corals experienced less stress with the decrease of excessive photosynthetic activities [65], indicating that light might be a key contributor to deep coral refuges [29]. In this context, recent literature suggested that ultraviolet radiation might be more important than PAR alone to explain bleaching [67–69]. While challenging to undertake such studies over large depth ranges [70], future work should include repeated monitoring surveys and endeavour to integrate additional environmental and biological parameters to further understand the observed decrease in bleaching over depth.

Mesophotic coral ecosystems are unlikely to provide long-term refugia from the ever-increasing frequency of thermal stress events [15,20]. However, they may still play a role as short-term refuges from bleaching. Our study has contributed to this still controversial concept by extending the depth-related decrease in bleaching incidence to lower mesophotic depths and demonstrating that bleaching is less prevalent towards the lower end of the depth range of coral taxa. While recognizing the geographical limitations of this study, and acknowledging the variability in bleaching susceptibility among genera, species and regions, the dampening impact of bleaching over depth may nonetheless be critical in preventing local species extinctions and safeguarding the unique fauna associated with mesophotic depths in some locations.

Ethics. Our work was done through the DEEPCORAL (ref. no. 3111) and the DEEPHOPE (ref. no. 168722) programmes. The present work consists of photo-quadrats (pictures) and environmental parameters only, and no export permits apply on French Polynesia for these data.

Data accessibility. Data and relevant code for this research work are stored in GitHub: https://github.com/gonzaloprb/ Deep_Bleaching_French_Polynesia and have been archived within the Zenodo repository: https://doi.org/10.5281/ zenodo.5211676.

Authors' contributions. G.P.-R., H.R. and L.H. planned the study. G.P-R., H.R. and UTP collected the data, G.P.-R. and V.P. analysed the data. G.P.-R., V.P. and G.T. wrote the first draft of the manuscript. All authors contributed to improve the study and the manuscript. All authors agree to be held accountable for the content herein and approve the final version of the manuscript.

Competing interests. The authors declare that there are no competing interests.

Funding. This research was funded by the ANR DEEPHOPE (ANRAAPG 2017 grant no. 168722), the Délégation à la Recherche DEEPCORAL, the CNRS DEEPREEF, the Agence Française pour la Biodiversité POLYAFB, the EPHE and the IFRECOR.

Acknowledgements. We sincerely thank the team of the Under The Pole Expedition III. Under The Pole Consortium: G. Bardout, J. Fauchet, A. Ferucci, F. Gazzola, G. Lagarrigue, J. Leblond, E. Marivint, A. Mittau, N. Mollon, N. Paulme, E. Périé-Bardout, R. Pete, S. Pujolle, G. Siu.

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
