## [Peer Review File · Royal Society Open Science]

Review History

RSOS-210139.R0 (Original submission)

Review form: Reviewer 1

Is the manuscript scientifically sound in its present form?

No

Are the interpretations and conclusions justified by the results?

No

Is the language acceptable?

Yes

Do you have any ethical concerns with this paper?

No

Have you any concerns about statistical analyses in this paper?

No

Recommendation?

Major revision is needed (please make suggestions in comments)

Comments to the Author(s)

My main issue with this manuscript is the numerous assumptions the authors made about how they interpreted their colony-level data. There were a number of items that were not fully described in the methods. First of all: the fact that the surveys were conducted 3-4 months after the bleaching event. The authors do not describe whether or not bleaching was ongoing during their surveys; the language suggests that it was not. The timing and length of the bleaching event needs to be clearly described. This length of time between the actual hot water event and surveys (3-4 mo) would allow for significant recovery in colonies that are less severely bleached (I.e., the "pale" category). And, obviously, this would greatly impact the scoring/classification of the colonies observed on the quadrats, which would then impact the data that informed the model.

The authors also did not describe how they determined "normal" coloration. This is an attribute that varies significantly within species with depth and can be differentially interpreted by different observers. They did not describe how they attributed "recent mortality" to bleaching, other than by stating that mortality due to bleaching was assumed (it should not be). What constituted recent mortality (vs. old) was also not described. Finally, they did not include a description of how they handled colonies that were patchily pale or bleached; I.e., how did they score colonies that were not presenting a uniform response?

Considering that all of these factors would have significant effects on the data set they accumulated, they need to be addressed and properly described in the Methods section. As the manuscript is written now, there are too many questions regarding the interpretation of their quadrat data to be confident in the rigorosity of their data set. Hopefully, the authors can provide detailed explanations which will address these concerns.

Review form: Reviewer 2

Is the manuscript scientifically sound in its present form?

No

Are the interpretations and conclusions justified by the results?

No

Is the language acceptable?

Yes

Do you have any ethical concerns with this paper?

No

Have you any concerns about statistical analyses in this paper?

No

Recommendation?

Reject

Comments to the Author(s)

In this manuscript, the authors attempt to demonstrate that mesophotic reefs are a refuge during thermal stress. While the idea may seem quite laudable, the authors do not present the necessary data and parameters to draw clear and innovative conclusions and thus do not propose a comprehensive and usable study. As it is, UV values and the genus of Symbiodiniaceae present in the colonies collected at different depths are missing and the temperature values are erroneous. As it stands, I cannot accept this publication, which does not seem to me to be at the level of Royal Society Open Science.

The authors declare that “their data suggest that the reduced prevalence of bleaching with depth, especially from shallow to upper mesophotic depths (40m), had a stronger relation with the attenuation of irradiance than temperature” but they never consider UVs, corals are threatened by increase in sea surface temperature but also by the incident flux of UV radiation (Häder et al 2007) more than by the light intensity. This increase in the incident flux of ultra-violet radiation is due to the effects of global warming on the stratospheric circulation and to a greater water stratification (Watanabe et al 2011), leading to a deeper penetration of UVR in the water column (Vodacek et al 1997). The combined effects of UVR and temperature strongly affect corals (Courtial et al 2017, D’Croz & Maté 2002, Fitt and Warner 1995).

Corals exposed to a simultaneous increase in temperature and UV radiation had bleached more strongly than under temperature stress alone (Glynn et al 1993, Fitt and Warner 1995, Lesser 1996, D’Croz and Mate 2000, Lesser and Farrell 2004, Ferrier-Pagès et al 2007), suggesting that UV radiation is a factor that compounds the effects of temperature. D’Croz and Mate (2000) also observed better recovery of corals after bleaching when protected from UV radiation. With climate change and the increase in the frequency of bleaching events, D’Croz and Mate’s observations suggest that UV-exposed reefs will have more difficulty recovering between bleaching events and will therefore be more fragile in the face of a new temperature stress. So more than light intensity data, it seems essential to have UV radiation values according to sites and depths, to link it with bleaching, which is sorely missing in this study.

No data is available on the genus of Symbiodiniaceae associated with corals depending on the site and depth, the study is focused on the sensitivity of corals to bleaching but nothing is given on this aspect. However, it is now widely established that some Symbiodiniaceae, such as for example *Durusdinium*, found in so-called "extremophilic" corals, can adapt to large variations in temperature and turbidity, coral colonies associated with this genus are less sensitive to bleaching (Silverstein et al 2017). I can imagine that the authors probably published these Symbiodiniaceae data in another article but the present study is thus rendered far too simplistic study that does not take into account the essential parameters/factors that could play a significant role in this sensitivity to bleaching.

Finally, the authors indicate in the suppl. materials Fig.S2 that « Data from Moorea do not correspond with the bleaching survey period » and that they used data from August 2018 I understand that it can be difficult to obtain temperature variations at such depths, but considering for the interpretation of the data, temperature variations during a period (August 2018), which does not correspond to the bleaching period, lacks credibility for a study focused on the sensitivity to bleaching...

Review form: Reviewer 3 (Heather Spalding)

Is the manuscript scientifically sound in its present form?

Yes

Are the interpretations and conclusions justified by the results?

Yes

Is the language acceptable?

Yes

Do you have any ethical concerns with this paper?

No

Have you any concerns about statistical analyses in this paper?

No

Recommendation?

Accept with minor revision (please list in comments)

Comments to the Author(s)

Overall Paper Comments

This is a clear and well-written manuscript that provides a physiological perspective of the deep reef refugia hypothesis through reduced coral bleaching at mesophotic depths. There are a few minor comments that need to be addressed. In particular, the terms "light" and "irradiance" are used interchangeably in the text and figures; one term should be chosen and used consistently. Comments shown by line number based on the page numbers at the bottom of the manuscript pages

Abstract, page 1

35-37: Reword to read "While bleaching episodes significantly impact shallow corals, little is known about their impact on mesophotic coral communities."

Introduction, page 2

36: delete "major"

38: delete "down"

43: change "lower" to "deeper"

Materials and Methods, page 3

10: How was the placement of the quadrats at each depth determined (haphazard?)?

Approximately what area at each depth was sampled, and was it similar or different among depths?

24: replace "since" with "because"

Results, page 4

37+ It seems strange to me that the % surface irradiance is standardized to 6 m. Shouldn't this be calculated based on the irradiance right below the surface of the water (% subsurface irradiance) or from values measured from above the surface (% surface irradiance)?

47+ Some mesophotic communities have sparse coral colonies with increasing depth. The density of coral colonies with increasing depth in the quadrats was not clear, but may be important to consider. In the results, can you give a range of the number of coral colonies found in each quadrat per depth. While Figure 1 shows a high density of corals in the quadrats, it is unclear if this is representative.

Discussion, page 6

24: Reword "but was also able to show" to "but also showed"

Discussion, page 7

12: Change "oceanographical" to "oceanographic"

Decision letter (RSOS-210139.R0)

Dear Dr Perez-Rosales,

The Editors assigned to your paper RSOS-210139 "Mesophotic coral communities escape thermal coral bleaching in French Polynesia" have now received comments from reviewers and would like you to revise the paper in accordance with the reviewer comments and any comments from the Editors. Please note this decision does not guarantee eventual acceptance.

Please submit your revised manuscript and required files (see below) no later than 21 days from today's (ie 23-Jul-2021) date. Note: the ScholarOne system will 'lock' if submission of the revision is attempted 21 or more days after the deadline. If you do not think you will be able to meet this deadline please contact the editorial office immediately.

on behalf of Dr Melita Samoilys (Associate Editor) and Pete Smith (Subject Editor)
openscience@royalsociety.org

Subject Editor Comments to Autho (Professor Pete Smith):

The three reviews are very different. I am recommending a major revision. You should address the key concerns of reviewers 1 and 2, and also address the minor comments from reviewer 3. Reviewer 2 is the most critical, if you can address their concerns then publication can proceed.

The two key issues are the definitions on recording bleaching in the methods are not clear (reviewer 1) and the lack of reference to UV radiation that reviewer 2 raises which has potentially serious implications in terms of interpretation of the results. I note reviewer 1 only comments on the methods yet asks for a major revision.

We look forward to receiving your revision.

Reviewer comments to Author:

Reviewer: 1

Comments to the Author(s)

My main issue with this manuscript is the numerous assumptions the authors made about how they interpreted their colony-level data. There were a number of items that were not fully described in the methods. First of all: the fact that the surveys were conducted 3-4 months after the bleaching event. The authors do not describe whether or not bleaching was ongoing during their surveys; the language suggests that it was not. The timing and length of the bleaching event needs to be clearly described. This length of time between the actual hot water event and surveys (3-4 mo) would allow for significant recovery in colonies that are less severely bleached (I.e., the "pale" category). And, obviously, this would greatly impact the scoring/classification of the colonies observed on the quadrats, which would then impact the data that informed the model.

The authors also did not describe how they determined "normal" coloration. This is an attribute that varies significantly within species with depth and can be differentially interpreted by different observers. They did not describe how they attributed "recent mortality" to bleaching, other than by stating that mortality due to bleaching was assumed (it should not be). What constituted recent mortality (vs. old) was also not described. Finally, they did not include a description of how they handled colonies that were patchily pale or bleached; I.e., how did they score colonies that were not presenting a uniform response?

Considering that all of these factors would have significant effects on the data set they accumulated, they need to be addressed and properly described in the Methods section. As the manuscript is written now, there are too many questions regarding the interpretation of their quadrat data to be confident in the rigorosity of their data set. Hopefully, the authors can provide detailed explanations which will address these concerns.

Reviewer: 2

Comments to the Author(s)

Please see attached file, "review Pérez-Rosales et al.pdf":

In this manuscript, the authors attempt to demonstrate that mesophotic reefs are a refuge during thermal stress. While the idea may seem quite laudable, the authors do not present the necessary data and parameters to draw clear and innovative conclusions and thus do not propose a comprehensive and usable study. As it is, UV values and the genus of Symbiodiniaceae present in the colonies collected at different depths are missing and the temperature values are erroneous. As it stands, I cannot accept this publication, which does not seem to me to be at the level of Royal Society Open Science.

The authors declare that "their data suggest that the reduced prevalence of bleaching with depth, especially from shallow to upper mesophotic depths (40m), had a stronger relation with the attenuation of irradiance than temperature" but they never consider UVs, corals are threatened by increase in sea surface temperature but also by the incident flux of UV radiation (Häder et al 2007) more than by the light intensity. This increase in the incident flux of ultra-violet radiation is due to the effects of global warming on the stratospheric circulation and to a greater water

stratification (Watanabe et al 2011), leading to a deeper penetration of UVR in the water column (Vodacek et al 1997). The combined effects of UVR and temperature strongly affect corals (Courtial et al 2017, D'Croz & Maté 2002, Fitt and Warner 1995).

Corals exposed to a simultaneous increase in temperature and UV radiation had bleached more strongly than under temperature stress alone (Glynn et al 1993, Fitt and Warner 1995, Lesser 1996, D'Croz and Mate 2000, Lesser and Farrell 2004, Ferrier-Pagès et al 2007), suggesting that UV radiation is a factor that compounds the effects of temperature. D'Croz and Mate (2000) also observed better recovery of corals after bleaching when protected from UV radiation. With climate change and the increase in the frequency of bleaching events, D'Croz and Mate's observations suggest that UV-exposed reefs will have more difficulty recovering between bleaching events and will therefore be more fragile in the face of a new temperature stress.

So more than light intensity data, it seems essential to have UV radiation values according to sites and depths, to link it with bleaching, which is sorely missing in this study.

No data is available on the genus of Symbiodiniaceae associated with corals depending on the site and depth, the study is focused on the sensitivity of corals to bleaching but nothing is given on this aspect. However, it is now widely established that some Symbiodiniaceae, such as for example *Durusdinium*, found in so-called "extremophilic" corals, can adapt to large variations in temperature and turbidity, coral colonies associated with this genus are less sensitive to bleaching (Silverstein et al 2017). I can imagine that the authors probably published these Symbiodiniaceae data in another article but the present study is thus rendered far too simplistic study that does not take into account the essential parameters/factors that could play a significant role in this sensitivity to bleaching.

Finally, the authors indicate in the suppl. materials Fig.S2 that « Data from Moorea do not correspond with the bleaching survey period » and that they used data from August 2018 I understand that it can be difficult to obtain temperature variations at such depths, but considering for the interpretation of the data, temperature variations during a period (August 2018), which does not correspond to the bleaching period, lacks credibility for a study focused on the sensitivity to bleaching...

Reviewer: 3

Comments to the Author(s)

Please see attached file, "Review_comments.docx".

Overall Paper Comments

This is a clear and well-written manuscript that provides a physiological perspective of the deep reef refugia hypothesis through reduced coral bleaching at mesophotic depths. There are a few minor comments that need to be addressed. In particular, the terms "light" and "irradiance" are used interchangeably in the text and figures; one term should be chosen and used consistently. Comments shown by line number based on the page numbers at the bottom of the manuscript pages

Abstract, page 1

35-37: Reword to read "While bleaching episodes significantly impact shallow corals, little is known about their impact on mesophotic coral communities."

Introduction, page 2

36: delete "major"

38: delete "down"

43: change "lower" to "deeper"

Materials and Methods, page 3

10: How was the placement of the quadrats at each depth determined (haphazard?)? Approximately what area at each depth was sampled, and was it similar or different among depths?

24: replace "since" with "because"

Results, page 4

37+ It seems strange to me that the % surface irradiance is standardized to 6 m. Shouldn't this be calculated based on the irradiance right below the surface of the water (% subsurface irradiance) or from values measured from above the surface (% surface irradiance)?

47+ Some mesophotic communities have sparse coral colonies with increasing depth. The density of coral colonies with increasing depth in the quadrats was not clear, but may be important to consider. In the results, can you give a range of the number of coral colonies found in each quadrat per depth. While Figure 1 shows a high density of corals in the quadrats, it is unclear if this is representative.

Discussion, page 6

24: Reword "but was also able to show" to "but also showed"

Discussion, page 7

12: Change "oceanographical" to "oceanographic"

===PREPARING YOUR MANUSCRIPT===

===PREPARING YOUR REVISION IN SCHOLARONE===

To revise your manuscript, log into <https://mc.manuscriptcentral.com/rsos> and enter your Author Centre - this may be accessed by clicking on "Author" in the dark toolbar at the top of the

page (just below the journal name). You will find your manuscript listed under "Manuscripts with Decisions". Under "Actions", click on "Create a Revision".

Author's Response to Decision Letter for (RSOS-210139.R0)

See Appendix A.

RSOS-210139.R1

Review form: Reviewer 2

Is the manuscript scientifically sound in its present form?

No

Are the interpretations and conclusions justified by the results?

No

Is the language acceptable?

Yes

Do you have any ethical concerns with this paper?

No

Have you any concerns about statistical analyses in this paper?

No

Recommendation?

Reject

Comments to the Author(s)

The authors did not add any of the essential data/information requested in the previous review. I stand by the criticism/advice given in my previous review. I am not more convinced by the innovative character of this work which in my opinion is not at the level of your journal. I can only maintain my position concerning this article.

Decision letter (RSOS-210139.R1)

Dear Dr Perez-Rosales,

It is a pleasure to accept your manuscript entitled "Mesophotic coral communities escape thermal coral bleaching in French Polynesia" in its current form for publication in Royal Society Open Science. The comments of the reviewer(s) who reviewed your manuscript are included at the foot of this letter.

on behalf of Dr Melita Samoilys (Associate Editor) and Pete Smith (Subject Editor)
openscience@royalsociety.org

Associate Editor Comments to Author (Dr Melita Samoilys):

Comments to the Author:

The authors have responded to the reviewers' comments well and the manuscript is ready to go to publication.

Namely, for reviewer 1 authors have justified clearly and adequately why a snapshot in time for measuring bleaching ~2 months after Max DHW is relevant if all sites were surveyed at that time in the same way and the hypothesis being tested referred to level of bleaching versus depth. They have also clearly addressed many other comments on the materials and methods which has greatly improved this section. These revisions to the text now greatly improve the manuscript; the reviewer has been very helpful.

Regarding reviewer No 2's point that the work was insufficient because it missed data on UV radiation and quantitative assessment of Symbiodiniaceae communities. I believe the authors have explained clearly why this was not necessary in their particular piece of research which sought simply to quantify levels of bleaching across a large depth gradient in a new location in French Polynesia. This remains a valuable contribution to these lesser known ecosystems worthy of publication. They also highlight this gap in their Discussion and recommend future work attempts to record UV radiation as well as PAR (Page 8 Lines 32-41). I believe this is adequate.

The authors have made many revisions which have greatly improved the manuscript and therefore my recommendation is to Accept As Is.

Reviewer comments to Author:

Reviewer: 2

Comments to the Author(s)

The authors did not add any of the essential data/information requested in the previous review. I stand by the criticism/advice given in my previous review. I am not more convinced by the innovative character of this work which in my opinion is not at the level of your journal. I can only maintain my position concerning this article.

Appendix A

Major revision of Manuscript ID RSOS-210139

Title: Mesophotic coral communities escape thermal coral bleaching in French Polynesia

Subject Editor Comments to Author (Professor Pete Smith):

The three reviews are very different. I am recommending a major revision. You should address the key concerns of reviewers 1 and 2, and also address the minor comments from reviewer 3. Reviewer 2 is the most critical, if you can address their concerns then publication can proceed. The two key issues are the definitions on recording bleaching in the methods are not clear (reviewer 1) and the lack of reference to UV radiation that reviewer 2 raises which has potentially serious implications in terms of interpretation of the results. I note reviewer 1 only comments on the methods yet asks for a major revision. We look forward to receiving your revision.

Dear Professor Pete Smith,

We would like to thank you for your time and efforts in handling our manuscript, “*Mesophotic coral communities escape thermal coral bleaching in French Polynesia*”. We would like to also thank the reviewers for their feedback, concerns and constructive suggestions, which we believe have improved our manuscript. We have now fully addressed their concerns and modified our manuscript considering the raised issues.

For ease of editorial review, we have included the Reviewers’ comments below in black, and our responses are in *italics* and blue.

Reviewer comments to Author:

Reviewer 1:

My main issue with this manuscript is the numerous assumptions the authors made about how they interpreted their colony-level data. There were a number of items that were not fully described in the methods.

First of all: the fact that the surveys were conducted 3-4 months after the bleaching event. The authors do not describe whether or not bleaching was ongoing during their surveys; the language suggests that it was not. The timing and length of the bleaching event needs to be clearly described. This length of time between the actual hot water event and surveys (3-4 mo) would allow for significant recovery in colonies that are less severely bleached (I.e., the "pale" category). And, obviously, this would greatly impact the scoring/classification of the colonies observed on the quadrats, which would then impact the data that informed the model.

We apologise if the submitted manuscript was not sufficiently clear in describing whether bleaching was recorded during the survey. We have now modified our manuscript to provide details as to when we performed the surveys, and we specified the potential impacts our monitoring dates can have on collected data. Our surveys (i.e., June-July) were performed 2-3 months after the peak of Degree Heating Weeks (i.e., end of March, mid-April), based on the posterior study of sea surface temperatures that revealed bleaching (see Fig. 1 B). In the figures and in our manuscript, we presented the timing and length of the bleaching event, which displays the time when Satellite Sea Surface Temperatures were above the bleaching threshold and the cumulative degree heating weeks of the whole bleaching episode.

The maximum signs of bleaching on colonies were at the very end of April and May just after the peak of the heat stress.

Satellite sea surface temperatures decreased below the bleaching thresholds during the surveys, so it is possible that some of the colonies we classified as "pale" resulted from an already occurring recovery (from "bleached" during just after the peak of heat stress to "pale" during the survey). Similarly, some colonies classified as "healthy" might have recovered from "pale" or "bleached" at the time of the surveys. As our monitoring over such a wide depth range was performed at a single snapshot of time, we cannot estimate

the percentages of maximum bleaching. However, we believe that if colonies recovered in less than few weeks after the heat stress peak, it reveals that colonies were resistant. Conversely, colonies that suffered from heat stress and were still bleached at the monitoring will remain bleached for weeks before the symbionts are back or ultimately die [1]. Therefore, the monitoring performed in June (Makatea) and July (Moorea) accounted for the maximum impacts of the bleaching event. In addition, since the objective of our study was to test the hypothesis that coral bleaching impacts were less severe in deeper depths than in shallow waters, the critical point was to perform all analyses at each site at the same time. Although monitoring did not occur during the peak of the bleaching, the present data still reflects the bleaching intensity along the depth gradient. Indeed, we agree that the timing matters when surveying the impacts of coral bleaching [2]; however, we believe that if multiple monitoring is not possible, collecting data on coral health to assess coral bleaching is more pertinent a few months after the peak of DHW to reveal the severity of bleaching clearly [2]. In this regard, we provided the data (e.g., exact dates of sampling, the peak of heat stress, accumulated degree heating weeks of the whole bleaching episode) required according to the available literature [2].

References:

- 1. Sakai K, Singh T, Iguchi A. 2019 Bleaching and post-bleaching mortality of Acropora corals on a heat-susceptible reef in 2016. Peer J. 7:e8138. (doi: 10.7717/peerj.8138)*
- 2. Claar DC, Baum JK. 2018 Timing matters: survey timing during extended heat stress can influence perceptions of coral susceptibility to bleaching. Coral Reefs 2018 38, 559–565. (doi:10.1007/S00338-018-01756-7)*

Additionally, complementary monitoring (i.e., April-May and October-November) in several shallow reefs of French Polynesia (data still under analysis) confirmed that colonies remained bleached and the severely impacted died for the ongoing six to 12 months, despite that temperatures decreased below heat stress.

Although we agree that having multiple time points for monitoring bleaching before, during and after would have been more powerful, it is to be remembered that the study and exploration of mesophotic depths are more difficult than on shallow reefs. Despite coming from a single snapshot of time, we believe that the present data is enough to test and respond to our hypothesis. Overall, our goal was not to describe the evolution of

colonies during a bleaching event. Instead, to show how the depth can escape the consequences of a thermal bleaching episode.

To make this more explicit, we emphasised that our bleaching survey consisted of single monitoring (snapshot of time) and that our results apply to the state of the colonies at the moment of the monitoring, 2-3 months after the peak of heat stress. We have now added this information in different sections of the manuscript.

- *ABSTRACT Line 4: “We studied the prevalence of coral bleaching, 2-3 months after the heat stress, along an extreme depth range from 6 to 90 m in French Polynesia.”*
- *RESULTS – “Bleaching along the depth gradient” Line 5: “Consistently, our model shows that the overall probability of bleaching across all coral genera, or dying shortly before the survey period, presumably due to bleaching, decreases with depth (Fig. 1B).”*
- *DISCUSSION Line 2: “This study shows a decrease in the incidence of coral bleaching across an extreme depth range (i.e. 6 to 90 m) in French Polynesia, as measured 2-3 months after the peak of a heat stress event.”*

The authors also did not describe how they determined "normal" coloration. This is an attribute that varies significantly within species with depth and can be differentially interpreted by different observers.

We agree with the reviewer that potential changes of colour might be the result of depth. However, we are confident in our categorization of bleaching levels with depth. Aside from the use of reference colour cards [3], we considered the normal colouration based on previous observations of healthy coral colonies with depth. The present analysis is following observation, collection and examination of 2,800 quadrats of mesophotic corals in French Polynesia [4]. The same observer performed all the photo-quadrats, trained to recognize healthy, pale, bleached and recently dead colonies. The use of a unique observer limits the error and variability in the health state scoring levels of corals and prevents potential bias in the interpretation.

References:

3. *Siebeck UE, Marshall NJ, Klüiter A, Hoegh-Guldberg O. 2006 Monitoring coral bleaching using a colour reference card. Coral Reefs 25, 453–460. (doi:10.1007/s00338-006-0123-8)*

4. *Pichon M, Rouzé H, Pérez-Rosales G & Hédouin L. 2021 Deep diving in paradise shines new light on the twilight zone: Preliminary results of the “Deephope” mesophotic programme in French Polynesia. 14th Internat. Coral Reef Symp. Bremen, Germany. Abstract ICRS2021-1731; Lives stream session 6B-C*

We have now specified these points in the MATERIALS AND METHODS, “Coral bleaching classification and statistical analysis”.

- *Line 5: “Although natural changes of colouration can vary with depth, we considered as references the previous healthy colours observed in the photo-quadrats of the DEEPHOPE expedition [31]”*
- *Line 17: “To prevent potential observer bias, all photo-quadrats were analysed and scored by a single observer who had been previously trained with the analysis of 2,880 quadrats from the DEEPHOPE expedition [31].”*

They did not describe how they attributed "recent mortality" to bleaching, other than by stating that mortality due to bleaching was assumed (it should not be). What constituted recent mortality (vs. old) was also not described.

We have now added detailed explanations of what differentiates old vs recent mortality. In short, colonies classified under “recently dead” are colonies with no living tissues remaining over the white skeleton and from which we can easily discern skeletal structure even if starting to be covered by turfing algae to varying extents. This category was clearly differentiated from old mortality, where the skeletal structure was invisible and totally overgrown by thick and dark turf, crustose coralline algae and other boring organisms. Colonies under the category old mortality were not included in our analysis to prevent bias due to mortality from previous disturbances.

We added these precisions in the MATERIALS AND METHODS, “Coral bleaching classification and statistical analysis section”

- *Line 12: “By contrast, colonies overgrown by thick and dark turf, crustose coralline algae and boring organisms were excluded from the analysis as their mortality likely predates the studied thermal coral bleaching event.”*

We understand the reviewer’s concerns regarding the second point “mortality due to bleaching was assumed (it should not be)”. Since we did not follow bleaching at the

individual levels before, during and after bleaching, we had to assume that recent mortality “recently dead” were colonies that died from the heat stress. We specified this in the manuscript “dying shortly before the survey period presumably due to bleaching.” Nonetheless, most papers published on coral bleaching and mortality assume that after a heat stress event, the recent mortality is due to the thermal bleaching episode [5-6]. Based on the number of works published with this assumption, we are confident that the recent mortality observed after the heat stress in Polynesia, with no other disturbances observed, was due to bleaching.

References

5. *Hughes TP et al. 2018 Global warming transforms coral reef assemblages. Nature. 556(7702):492-6. (doi: 10.1038/s41586-018-0041-2)*
6. *Hédouin L et al. 2020 Contrasting patterns of mortality in Polynesian coral reefs following the third global coral bleaching event in 2016 Coral Reefs 39, 939–952. (doi:10.1007/s00338-020-01914-w)*

Finally, they did not include a description of how they handled colonies that were patchily pale or bleached; I.e., how did they score colonies that were not presenting a uniform response?

We apologize for this missing information. We have now expanded the MATERIALS AND METHODS, “Coral bleaching classification and statistical analysis section.”

- *Line 15: “Finally, when colonies were patchily pigmented, they were classified by the most severely bleached patch (e.g., a colony 25% bleached and 75% pale was considered as bleached and colonies 30% pale and 70% healthy as pale).”*

Considering that all of these factors would have significant effects on the data set they accumulated, they need to be addressed and properly described in the Methods section.

As the manuscript is written now, there are too many questions regarding the interpretation of their quadrat data to be confident in the rigorousness of their data set. Hopefully, the authors can provide detailed explanations which will address these concerns.

We thank the reviewer for her/his constructive comments and suggestions, which we have carefully addressed. We acknowledge that the first submitted version of the manuscript missed some detailed information. We have now added further explanations providing pertinent details to avoid interpretations of assumptions by the readers and also to provide evidence of the robustness and rigorousness of our data.

We want to finish by saying that although our methodology might still have some minor issues, all available ecological studies assessing coral bleaching have some at some points. In our particular study, we also deal with the challenges of difficult-to-study mesophotic depths. We designed our study to gather unique data from an unexplored environment (down to 90 m depth) during a heatwave that affected French Polynesia. Our general results provide novel and trustworthy information on an unprecedented worldwide depth range: the incidence of coral bleaching decreased along the depth gradient in our study sites of French Polynesia.

Reviewer 2:

In this manuscript, the authors attempt to demonstrate that mesophotic reefs are a refuge during thermal stress. While the idea may seem quite laudable, the authors do not present the necessary data and parameters to draw clear and innovative conclusions and thus do not propose a comprehensive and usable study. As it is, UV values and the genus of Symbiodiniaceae present in the colonies collected at different depths are missing and the temperature values are erroneous. As it stands, I cannot accept this publication, which does not seem to me to be at the level of Royal Society Open Science.

The authors declare that “their data suggest that the reduced prevalence of bleaching with depth, especially from shallow to upper mesophotic depths (40m), had a stronger relation with the attenuation of irradiance than temperature” but they never consider UVs, corals are threatened by increase in sea surface temperature but also by the incident flux of UV radiation (Häder et al 2007) more than by the light intensity. This increase in the incident flux of ultra-violet radiation is due to the effects of global warming on the stratospheric circulation and to a greater water stratification (Watanabe et al 2011), leading to a deeper penetration of UVR in the water column (Vodacek et al 1997). The combined effects of

UVR and temperature strongly affect corals (Courtial et al 2017, D’Croz & Maté 2002, Fitt and Warner 1995).

Corals exposed to a simultaneous increase in temperature and UV radiation had bleached more strongly than under temperature stress alone (Glynn et al 1993, Fitt and Warner 1995, Lesser 1996, D’Croz and Mate 2000, Lesser and Farrell 2004, Ferrier-Pagès et al 2007), suggesting that UV radiation is a factor that compounds the effects of temperature. D’Croz and Mate (2000) also observed better recovery of corals after bleaching when protected from UV radiation. With climate change and the increase in the frequency of bleaching events, D’Croz and Mate’s observations suggest that UV-exposed reefs will have more difficulty recovering between bleaching events and will therefore be more fragile in the face of a new temperature stress.

So more than light intensity data, it seems essential to have UV radiation values according to sites and depths, to link it with bleaching, which is sorely missing in this study.

No data is available on the genus of Symbiodiniaceae associated with corals depending on the site and depth, the study is focused on the sensitivity of corals to bleaching but nothing is given on this aspect. However, it is now widely established that some Symbiodiniaceae, such as for example *Durusdinium*, found in so-called "extremophilic" corals, can adapt to large variations in temperature and turbidity, coral colonies associated with this genus are less sensitive to bleaching (Silverstein et al 2017). I can imagine that the authors probably published these Symbiodiniaceae data in another article but the present study is thus rendered far too simplistic study that does not take into account the essential parameters/factors that could play a significant role in this sensitivity to bleaching.

We thank the reviewer for her/his interest in the aims of our study and comments. However, we believe that the reviewer asks for changes but seems not to consider relevant statements from our manuscript. We believe that the arguments provided by the reviewer are not mandatory to test the hypothesis of our work: to study if coral bleaching incidence decreases with depth. While we agree with the reviewer on the importance of the role of Symbiodiniaceae in bleaching susceptibility and the potential effects of UV radiation to understand the biological/physiological mechanisms of coral bleaching, we wish to stress that the aim of our work was different. We did not aim to understand why coral bleaching decreases with depth but to determine whether corals from mesophotic coral ecosystems

were threatened similarly to shallow reefs by thermal bleaching. Considering the crucial lack of data available on mesophotic reefs (i.e., we provide unprecedented depth range worldwide and the first bleaching assessment with depth in French Polynesia), the first scientific rationale is to test how thermal stress impacts coral bleaching with depth.

The reviewer seems to argue that our manuscript is not valuable for publication because we did not measure UV values or quantitative abundances of Symbiodiniaceae. However, a bibliography of the recent publications on coral bleaching with depth highlighted a vast literature of peer-reviewed papers in the most prestigious journals [1,2], on which UV measurements and/or abundance of Symbiodiniaceae communities are not considered at all.

References:

1. Stuart-Smith RD, Brown CJ, Ceccarelli DM, Edgar GJ Ecosystem restructuring along the Great Barrier Reef following mass coral bleaching 2018 *Nature* 560 (7716):92-6. (doi:10.1038/s41586-018-0359-9)
2. Hughes TP et al. 2018 Spatial and temporal patterns of mass bleaching of corals in the Anthropocene. *Science*. 5359(6371):80-3. (doi: 10.1126/science. aan8048)

Furthermore, if the reviewer feels that UV radiation are mandatory because we studied the effects of coral bleaching with depth, we would like to refer to the available literature on coral bleaching in mesophotic depths without any information nor mention of UV measurements.

- Muir PR et al. 2017 Species identity and depth predict bleaching severity in reef-building corals: shall the deep inherit the reef? *Proc. R. Soc. B Biol. Sci.* 284, 20171551. (doi:10.1098/rspb.2017.1551).
- Frade PR et al. 2018 Deep reefs of the Great Barrier Reef offer limited thermal refuge during mass coral bleaching. *Nat. Commun.* 9, 1–8. (doi:10.1038/s41467-018-05741-0)
- Baird A et al. 2018 A decline in bleaching suggests that depth can provide a refuge from global warming in most coral taxa. *Mar. Ecol. Prog. Ser.* 603, 257–264. (doi:10.3354/meps12732)

Although we believe that UV radiation may have helped to better understand the causes of bleaching, this type of data was not necessary to test our hypothesis. We believe that the request is not as mandatory as raised by the reviewer and the literature supports this.

Although we disagree that not presenting data of Symbiodiniaceaea impedes publishing on coral bleaching with depth, we agree that quantifying symbiotic communities would have allowed us to understand the mechanisms of why coral bleaching decreased with depth. However, and as mentioned above, this was not the goal of our paper. When authors target molecular analysis to document the change in Symbiodiniaceaea communities during a bleaching event, they focus on a few (one or two) species but never on the whole range of species present on the reef. In our study site, the coral community composition changes with depth. Some species are only present at 90 m and others at 6 m depth, so no comparison along the depth gradient would have been possible, except for the rare generalist coral species living at multiple depths. Therefore, if Symbiodiniaceaea communities were to be targeted during a bleaching event, the hypothesis and the design of the study would have been totally different from the objective of our study. Nevertheless, we agree with the reviewer on the interest of gathering symbiotic communities. Together with the bacteria, virus, and the whole microbiome, but also with other physiological processes, we would have probably managed to identify the mechanisms involved in bleaching resistance/susceptibility that explain why coral bleaching decreased with depth. Last but not least, the literature provided above for UV radiation does not account for the endosymbiotic symbionts either.

However, this was not the goal of our paper because, first of all, we did not even know whether coral bleaching occurred in-depth. Yet, we added a section raising the future interest of integrating additional parameters to understand the mechanisms of why coral bleaching decreased with depth. These are in the DISCUSSION “Page 8”

- *Line 15: “In this context, recent literature suggested that Ultra Violet (UV) radiation might be more important than PAR alone to explain bleaching [69]. While challenging to undertake such studies over large depth ranges [70], future work should include repeated monitoring surveys and endeavour to integrate additional environmental and biological parameters to further understand the observed decrease in bleaching over depth.”*

Finally, we still believe that our data is of critical interest for the ecological aim of our study. In other words, to study how/if coral bleaching decreases with depth rather than why. Based on our study and results, we demonstrated that thermal coral bleaching consequences decrease with depth. For the first time (a) over such a wide depth range (6-90 m) and (b) in French Polynesia.

Finally, the authors indicate in the suppl. materials Fig.S2 that « Data from Moorea do not correspond with the bleaching survey period » and that they used data from August 2018 I understand that it can be difficult to obtain temperature variations at such depths, but considering for the interpretation of the data, temperature variations during a period (August 2018), which does not correspond to the bleaching period, lacks credibility for a study focused on the sensitivity to bleaching...

We apologise, but we do not understand why our temperature values are declared “erroneous” at the beginning of the review. We displayed the temperature that induced bleaching derived from the accumulated heat stress calculated from satellite sea surface temperatures data (Fig. 1 B). These temperatures correspond with the bleaching episode that affected our colonies. In addition, we displayed the graph Fig. S2 to highlight the high variability of temperature with increasing depth (C). We mentioned in the manuscript that the logger temperatures were not recorded during the bleaching heat stress. For this reason, temperatures do not match as they were not recorded at the same time.

In order to clarify this issue, we replaced the legend of Fig. S2 from “Data from Moorea do not correspond with the bleaching survey period” with “In situ temperature and light data from Moorea was recorded in August 2018.” We also emphasized why such data are important (despite the fact that they were not recorded during the bleaching period). We were interested in the variability of temperature with depth, and this variability was constant at all study sites from August 2018 to July 2019. Moreover, we only provided the relation of bleaching with temperature loggers in supplementary figures, and we avoided concluding from it as it deserves further studies.

Finally, please believe that we know that the reviewers' remarks brought ambitious questions to the Mesophotic Research, and we also hope to see those challenging assessments tackled in the future. Although our study has room for improvement, as in our humble opinion all studies have, we believe that the data presented in our study are robust enough to support our conclusion. At least in the conditions and locations of our study, the effects of thermal coral bleaching decreased along the depth gradient. We believe that even in the absence of supporting environmental data, the reviewer completely disregards the novelty of gathering these bleaching data over such a wide depth range.

Reviewer 3:

Overall Paper Comments

This is a clear and well-written manuscript that provides a physiological perspective of the deep reef refugia hypothesis through reduced coral bleaching at mesophotic depths. There are a few minor comments that need to be addressed. In particular, the terms “light” and “irradiance” are used interchangeably in the text and figures; one term should be chosen and used consistently.

We thank the reviewer. We have now addressed the minor comments in our manuscript.

Comments shown by line number based on the page numbers at the bottom of the manuscript pages

Abstract, page 1

35-37: Reword to read “While beaching episodes significantly impact shallow corals, little is known about their impact on mesophotic coral communities.”

We have reworded as suggested.

Introduction, page 2

36: delete “major”

Done.

38: delete “down”

Done.

43: change “lower” to “deeper”

Done.

Materials and Methods, page 3

10: How was the placement of the quadrats at each depth determined (haphazard)? Approximately what area at each depth was sampled, and was it similar or different among depths?

We have addressed this as follows:

MATERIALS AND METHODS, “Study locations and sampling protocol”

- *Line 10: “Thirty photo-quadrats (0.75 x 0.75 m) were randomly taken at each of five different depths (6, 20, 40, 60 and 90 m; a total of 150 photos at each location), with constant sampling effort at each depth, covering around 17m² per isobath and 85m² per location (Fig. 1A).”*

24: replace “since” with “because”

Done.

Results, page 4

37+ It seems strange to me that the % surface irradiance is standardized to 6 m. Shouldn't this be calculated based on the irradiance right below the surface of the water (%subsurface irradiance) or from values measured from above the surface (% surface irradiance)?

We thank you for bringing this up. We have standardised irradiance to 6 m because it was the shallowest depth of our loggers and the shallowest depth on which we measured coral bleaching. Although closer to the surface would have been ideal, the hydrodynamics of waves and currents did not allow to place either loggers or photo-quadrat at such a depth.

Since we also quantified the light with a CTD light logger profile, we could see that the light reduction and relative values agreed with the vertical profile normalised to the surface S. Fig. 3.

47+ Some mesophotic communities have sparse coral colonies with increasing depth. The density of coral colonies with increasing depth in the quadrats was not clear, but may be important to consider. In the results, can you give a range of the number of coral colonies found in each quadrat per depth. While Figure 1 shows a high density of corals in the quadrats, it is unclear if this is representative.

We thank the reviewer for the suggestion. We have added a sentence to explain that the changes in the mean average in the number of coral colonies per quadrat at each depth differentiated between locations but did not significantly affect the proportions of coral bleaching with depth. RESULTS, “Bleaching along the depth gradient”

- *Line 2: “Despite differences between the two locations in the numbers of colonies over depth (e.g., decreasing in Moorea below 40 m, and remaining constant in Makatea with a maximum number at 60 m), they did not significantly affect the proportional measures of coral bleaching with depth.”*

Discussion, page 6

24: Reword “but was also able to show” to “but also showed”

Done.

Discussion, page 7

12: Change “oceanographical” to “oceanographic”

Done.

On behalf of all authors, we want to thank the editors and reviewers for taking the time and helping us improve our manuscript.